# Soluble triggering receptor expressed on myeloid cells 1 is associated with hemoconcentration and endothelial activation in children and young adults with dengue virus infection in the Philippines

Jiayi Yang[1], Hridesh Mishra[2], Michelle Ngai[2], Vanessa Tran[2], Maria Salome Siose Painaga[3], James Yared Gaite[3], Ashley Roberts[1], Kevin C. Kain[4,5☯], Michael T. Hawkes[1☯]*

1 Department of Pediatrics, University of British Columbia, Vancouver, Canada, 2 SAR Laboratories, Sandra Rotman Centre for Global Health, University Health Network-Toronto General Hospital Research Institute, Toronto, Canada, 3 Department of Infectious Diseases, Lebumfacil-Santa Ana Medical Center, Toledo, Cebu, Philippines, 4 Department of Laboratory Medicine and Pathobiology, Faculty of Medicine, University of Toronto, Toronto, Canada, 5 Tropical Disease Unit, Division of Infectious Diseases, Department of Medicine, UHN-Toronto General Hospital, Toronto, Canada

☯ Contributed equally to this work.
* m.t.hawkes@ubc.ca

## Abstract

### Background

Triggering receptor expressed on myeloid cells 1 (TREM1) is a cell-surface receptor expressed on neutrophils that amplifies the inflammatory response. Dengue virus (DENV) infection is characterized by systemic inflammation, endothelial activation, and vascular leakage.

### Methodology/Principal Findings

We investigated circulating soluble TREM-1 (sTREM-1) levels in 244 children and young adults aged 1–26 years with dengue fever presenting to an outpatient clinic in the Philippines. Elevated sTREM-1 (≥130 pg/mL) was associated with hemoconcentration, a hallmark of vascular leakage (odds ratio (OR) 3.8, 95%CI 1.6-10, p = 0.0020). In turn, hemoconcentration was associated with hospitalization (OR 4.2, 95%CI 1.0-38, p = 0.0497) and higher volume of intravenous fluid required for resuscitation (p = 0.019). Elevated inflammation marker TNF (≥5 pg/mL) was associated with increased sTREM-1 levels (p = 0.0014). Endothelial activation markers angiopoietin-2 (Ang-2), soluble FMS-like tyrosine kinase-1 (sFlt-1), and soluble vascular cell adhesion molecule 1 (sVCAM-1) were correlated with sTREM-1 levels (p < 0.0001 for all three comparisons).

**Data availability statement:** The authors confirm that all data underlying the findings are fully available without restriction. All relevant data are within the paper and its Supporting Information files.

**Funding:** This study was supported by the Tesari Foundation, Canadian Institutes of Health Research (CIHR) Foundation grant FDN-148439 (KCK), Canada Research Chair (KCK), and kind donations from Kim Kertland, and Rotary International (K-W group). JY was funded through a summer studentship F24-01744 from the Healthy Starts pillar of the British Columbia Children's Health Research Institute. The funders had no role in study design, data collection, data analysis, data interpretation, writing of the report, or decision to submit the article for publication.

**Competing interests:** The authors have declared that no competing interests exist.

### Conclusions/significance

Our findings suggest that sTREM-1 may be a clinically informative marker of neutrophil activation, associated with hemoconcentration, systemic inflammation, and endothelial activation and in dengue fever.

### Author summary

Dengue fever is a viral infection that can cause blood vessel leakage, leading to low blood pressure and shock. One sign of blood vessel leakage is hemoconcentration, whereby red blood cells becomes more concentrated in the blood stream. Soluble triggering receptor expressed on myeloid cell (sTREM-1) is a molecule on the surface of neutrophils (a type of white blood cell) that can be released into the circulation upon neutrophil activation. To investigate the relationship between sTREM-1 and vessel leakage, we conducted a prospective study of children and young adults with dengue fever in the Philippines. Elevated sTREM-1 level at the time of presentation to an outpatient clinic was associated with hemoconcentration. In turn, hemoconcentration was associated with a higher likelihood of hospitalization and greater need for intravenous fluid resuscitation. Patients with higher sTREM-1 levels also had elevated levels of the pro-inflammatory cytokine TNF, as well as elevated markers of endothelial activation (Ang2, sFlt1, and sVCAM-1). Our study highlights a potential role for sTREM-1 as a clinically informative marker to identify dengue patients at risk of vessel leakage.

### Introduction

Dengue fever (DF) is a tropical mosquito-borne viral infection [1]. Dengue incidence has surged 10-fold over the past 20 years, making it the fastest spreading arboviral disease [2]. The Philippines is a high-incidence country, with over 400,000 cases reported in 2019 [3].

The 2009 World Health Organization (WHO) guidelines classify DF into dengue ± warnings signs and severe dengue [4]. Warning signs (abdominal pain or tenderness, persistent vomiting, clinical fluid accumulation, mucosal bleeding, lethargy or restlessness, liver enlargement, and hemoconcentration with rapid decrease in platelet count) may herald progression to severe disease and warrant hospitalization for observation of vital signs [4]. Severe dengue includes severe plasma leakage (including dengue shock syndrome), severe bleeding (dengue hemorrhagic fever), and severe organ impairment [4]. DF is a dynamic disease which progresses through three phases: febrile, critical, and recovery [4]. The clinical evolution and outcome may be difficult to predict and prognostic tools are urgently needed for accurate risk stratification. While most patients recover without complications, a subset develops severe dengue and is at risk of adverse outcomes including death.

Hemoconcentration, a hallmark of vascular permeability, may precede dengue shock syndrome [4]. A meta-analysis of 143 studies identified increased hematocrit with a concurrent decrease in platelet count as a significant risk factor for severe dengue [5]. A pediatric study from the Philippines found that the presence of hemoconcentration was associated with a sixfold increase in dengue mortality [6].

Neutrophils are increasingly recognized for their role in DENV-induced plasma leakage. Activated neutrophils release cytokines, surface molecules, and neutrophil extracellular traps, potentially contributing to endothelial disruption, leading to plasma extravasation [7]. Increased circulating markers of neutrophil activation have been reported in severe dengue compared to uncomplicated cases [8].

Triggering Receptor Expressed on Myeloid Cells Type 1 (TREM-1) is a transmembrane protein found on peripheral neutrophils, monocytes, and macrophages [9]. TREM-1-ligand interactions initiate a signaling cascade that culminates in the release of inflammatory mediators. TREM-1 may be released from the cell surface into the circulation through proteolytic cleavage, resulting in a soluble form (sTREM-1) [10]. Elevated sTREM-1 concentrations indicate neutrophil activation and are associated with disease severity in sepsis and pneumonia [11,12]. DENV increases TREM-1 expression on cultured human neutrophils *in vitro* [13]. Early in the course of DENV infection, there is a transient decrease in surface expression of TREM-1 on neutrophils coinciding with increased sTREM-1, suggesting shedding of TREM-1 into the circulation [14]. A multi-center study identified sTREM-1 as one of a combination of inflammatory and vascular markers that predicted the development of moderate or severe dengue [15]. Taken together, these findings implicate TREM-1 in the innate inflammatory response to DENV and suggest that sTREM-1 may be a clinically informative marker of disease severity.

Here, we investigated plasma sTREM-1 levels in children and young adults with DENV disease at presentation to an outpatient clinic in the Philippines. Our primary objective was to investigate the association between sTREM-1 and hemoconcentration, a sign of vascular leakage. Secondary objectives were to examine the relationship between sTREM-1 and leukocyte count, inflammation, and markers of endothelial activation.

## Methods

### Ethics statement

The study was approved by the Institutional Review Board (IRB) of Chong Hua Hospital, Cebu City, Philippines, and the University Health Network Research Ethics Committee, Toronto, Canada (UHN REB Number 13–6168-AE). Written informed consent was obtained from participants or, in the case of children, from the parent or guardian.

### Study design and participants

A prospective cohort study of 244 patients was conducted at the Lebumfacil-Santa Ana Medical Centre in Toledo, Cebu province, Philippines. We previously reported the levels of angiopoietins in this cohort [15]. Patients were eligible for inclusion if they were 1 to <26 years old with clinically suspected DENV infection, confirmed through detection of the NS1 antigen or IgM using a rapid bedside test (Bioline DENGUE DUO, Abbott, Abbott Park, Illinois). Exclusion criteria were: (a) alternative bacterial/viral infections; (b) significant comorbidity; (b) pregnant or breastfeeding status; (d) recent blood transfusion within the past 3 months; and (e) prior study participation.

### Study procedures

Patients presenting to the outpatient clinic for clinical care were enrolled if eligible and provided informed consent. At this initial study visit, clinical characteristics were recorded and venipuncture blood was collected. The day of onset of symptoms was ascertained based on patient or parental report at this initial study visit. The WHO disease severity classification was assessed at this initial study visit based on presenting signs and symptoms.

The venipuncture sample from this initial visit was used for the quantification of plasma proteins. Concentrations of sTREM-1, tumor necrosis factor (TNF), angiopoietin-2 (Ang-2), soluble FMS-like tyrosine kinase-1 (sFlt-1), and vascular cell adhesion molecule 1 (sVCAM-1) were measured using a multiplex Luminex assay with custom reagents (R&D Systems, Minneapolis, MN). Protein concentrations was below the lower limit of detection of the assay were arbitrarily assigned a value equal to the lower limit of the assay. For sTREM-1, the lower limit of detection was 45 pg/mL. For TNF, the lower limit of detection was 5 pg/mL.

Over the following 14–21 days, patients were either referred to hospital, or managed as outpatients. A complete blood count (CBC) was obtained at the initial study visit and was repeated during the acute illness to monitor hematologic parameters, as often as clinically indicated. The timing of the repeat CBC assessments was according to the clinician's judgement. A final CBC was also obtained at a scheduled follow-up visit 14–21 days after study enrolment. To estimate the baseline hematocrit (HCT) for each patient, we used the value measured after recovery from the acute illness (day 14–21 after enrolment). Hemoconcentration was defined as a ≥20% rise in HCT relative to the baseline value, considering all available HCT measurements over the course of the illness [4].

Participants returned to the outpatient clinic in follow-up after resolution of their acute illness (day 14–21 after presentation) to review and document their clinical course (hemoconcentration, hospitalization, and intravenous fluid requirement).

## Statistical analysis

Summary statistics for categorical variables were reported as frequency (n) and percentage (%). Continuous variables were reported as median and interquartile range (IQR). Associations between categorical variables were examined using Pearson chi-squared test, or Fisher exact test, as appropriate. The comparisons of continuous variables employed the Wilcoxon rank sum test. Receiver operator characteristic (ROC) curve analysis was used to determine the optimal cutoff (Youden index) for sTREM-1 as a predictor of hemoconcentration. To verify the association between sTREM-1 and hemoconcentration, logistic regression models were constructed to adjust for potential confounding effects of clinical covariates. Subgroup analyses were performed, stratifying the cohort by age (below and above 9 years). Significance was assessed at the $\alpha = 0.05$ level. Data analysis and visualization were performed using GraphPad Prism version 6 (GraphPad Software Inc., La Jolla, CA, USA, 2012), and R (version 4.3.0).

## Sample size

We calculated that 201 patients were needed to detect a statistically significant difference in sTREM-1 levels between patients with and without hemoconcentration. This calculation assumed a mean (standard deviation) sTREM-1 level of 200 (150) pg/mL, a 50% difference in sTREM-1 between groups, and a proportion of 11% with hemoconcentration [14,16].

## Results

### Patient characteristics

A total of 244 children and young adults presenting to the outpatient clinic from September 2015 to February 2017 were included in the analysis. Based on the WHO classification [4], 63 patients (26%) were diagnosed as uncomplicated dengue without warning signs, 179 (73%) as dengue with warning signs, and 2 (<1%) as severe dengue. There were no fatalities. Patient characteristics at clinic presentation were summarized (Table 1).

The median time from symptom onset to presentation to the outpatient clinic was 3 days (IQR 2–4). A total of 181 (74%) of patients were hospitalized. The timing of hospital admission was on the same day as the initial clinic visit in 143 patients (82%), ranging from 0 to 3 days after presentation. The timing of the final follow-up study visit after recovery was a median of 15 days (IQR 14–15) after presentation and 18 days (IQR 17–20) after symptom onset. Clinical outcomes observed during follow-up were summarized (Table 2).

**Neglected Tropical Diseases** PLOS

**Table 1. Clinical and laboratory characteristics of dengue patients at clinic presentation.**

| | Overall (N = 244) | sTREM-1 < 130 pg/mL (N = 152) | sTREM-1 ≥ 130 pg/mL (N = 92) | *P* value |
|---|---|---|---|---|
| **Demographics** | | | | |
| Age (yr), median (IQR) | 9 (5-14) | 9 (5-14) | 8 (5-13) | 0.99 |
| Female sex, n (%) | 97 (40) | 69 (45) | 28 (30) | **0.029** |
| **Symptoms** | | | | |
| Fever (self-reported) | 244 (100) | 152 (100) | 92 (100) | >0.99 |
| Duration of fever (days) | 3 (2-4) | 3 (2-4) | 3 (3-5) | **0.042** |
| Rash | 89 (36) | 59 (39) | 30 (33) | 0.40 |
| Joint pain | 115 (47) | 65 (43) | 50 (54) | 0.10 |
| Abdominal pain | 126 (52) | 76 (50) | 50 (54) | 0.60 |
| Persistent vomiting (>2 days) | 79 (32) | 47 (31) | 32 (35) | 0.63 |
| Bleeding (nose, mouth, eyes, stool) | 33 (14) | 22 (14) | 11 (12) | 0.72 |
| **Physical Examination** | | | | |
| Weight (kg) | 23.3 (16.2-39.2) | 23.8 (16.2-39.2) | 23 (16.2-38.8) | 0.94 |
| Height (cm) | 127 (107-150) | 127 (105-150) | 126 (108-146) | 0.72 |
| Heart rate (/min) | 96 (86-110) | 97 (88-108) | 96 (84-112) | 0.47 |
| Tachycardia, n (%) | 24 (9.8) | 14 (9.2) | 10 (11) | 0.84 |
| Respiratory rate (/min) | 28 (24-32) | 28 (24-32) | 28 (26-32.5) | 0.73 |
| Tachypnea, n (%) | 197 (81) | 122 (80) | 75 (82) | 0.94 |
| Systolic blood pressure (mmHg) | 90 (80-100) | 90 (80-100) | 90 (80-100) | 0.42 |
| Diastolic blood pressure (mmHg) | 60 (50-60) | 60 (50-70) | 60 (50-60) | 0.061 |
| Hypotension, n (%) | 47 (19) | 26 (17) | 21 (23) | 0.35 |
| Positive tourniquet test | 45 (18) | 24 (16) | 21 (23) | 0.23 |
| Severity | | | | 0.11 |
| Uncomplicated | 63 (26) | 43 (28) | 20 (22) | |
| Warning signs | 179 (73) | 109 (72) | 70 (76) | |
| Severe dengue | 2 (0.82) | 0 (0) | 2 (2.2) | |
| Fatal | 0 | 0 | 0 | >0.99 |
| **Hematologic parameters** | | | | |
| Hemoglobin (g/L) | 132 (125-144) | 132 (124-142) | 135 (127-147) | 0.070 |
| Hematocrit (%) | 0.40 (0.38-0.43) | 0.40 (0.37-0.43) | 0.40 (0.38-0.45) | **0.043** |
| Platelet count (×10⁹/L) | 204 (144-260) | 204 (159-257) | 206 (118-269) | 0.20 |
| WBC (×10⁹/L) | 5.0 (2.8-6.0) | 4.5 (2.7-5.9) | 5.4 (3.4-6.3) | **0.0045** |
| Neutrophil count (×10⁹/L) | 2.1 (1.4-3.1) | 1.8 (1.2-2.9) | 2.3 (1.6-3.4) | **0.0078** |
| **Markers of endothelial activation** | | | | |
| Ang-2 (ng/mL) | 2.9 (2.1-4.2) | 2.7 (2.0-4.1) | 3.4 (2.5-4.8) | **0.00078** |
| sFlt1 (pg/mL) | 450 (260-700) | 398 (225-635) | 580 (383-762) | **<0.0001** |
| sVCAM-1 (mg/mL) | 1.7 (1.0-3.4) | 1.5 (1.0-2.9) | 2.7 (1.1-4.7) | **0.0096** |

### Elevated sTREM-1 is associated with hemoconcentration

Our primary objective was to examine the association between sTREM-1 and hemoconcentration. The median plasma sTREM-1 level at presentation was 56 pg/mL (IQR 45–145 pg/mL). Of note, 122 patients (50%) had sTREM-1 plasma concentration of <45 pg/mL (below the limit of detection of the assay) and were arbitrarily assigned a value of 45 pg/mL for subsequent analysis.

**Table 2. Clinical outcomes of dengue patients at follow-ups.**

| Clinical Outcomes | Overall (N==244) | sTREM-1 < 130 pg/mL (N=152) | sTREM-1 ≥ 130 pg/mL (N=92) | P value |
|---|---|---|---|---|
| Hemoconcentration, n (%) | 27 (11) | 9 (6.3) | 18 (20) | 0.0020 |
| Hospitalization, n (%) | 181 (74) | 110 (72) | 71 (77) | 0.50 |
| Intravenous fluid resuscitation (L) | 1.5 (0.0-4.0) | 1.5 (0.0-4.0) | 1.8 (0.0-4.0) | 0.70 |

Twenty-seven patients (11%) had evidence of hemoconcentration. The timing of the maximum hemoglobin level was on the day of presentation in all 27 patients. The HCT decreased thereafter with resolution of the infection. No patients developed hemoconcentration *de novo* after the initial clinic visit. Thus, the HCT measurement defining hemoconcentration was simultaneous with the sTREM-1 measurement and the association between sTREM-1.

Median sTREM-1 levels were significantly higher in patients with hemoconcentration (150 pg/mL versus 45 pg/mL, p=0.0045). Using ROC curve analysis, we determined that the optimal cutoff of sTREM-1 to discriminate between of hemoconcentration was 130 pg/mL (Fig 1A). Patients with elevated sTREM-1 (≥130 pg/mL) had a 3.8-fold higher odds (95%CI 1.6-10) of hemoconcentration (P=0.0020, Fig 1B). In turn, patients with hemoconcentration had a 4.2-fold higher odds (95%CI 1.0-38) of hospitalization, (P=0.0497, Fig 1C) and required a greater total volume of intravenous fluid for management (P=0.019, Fig 1D).

To adjust for potential confounding variables, we constructed logistic regression models of hemoconcentration as a function of sTREM-1 (continuous variable) and clinical covariates. Of note, biological sex and time from symptom onset were statistically significantly associated with sTREM-1 levels and were included as clinical covariates in the multivariable logistic regression model (Table 3). In univariable and multivariable models, sTREM-1 remained statistically significantly associated with hemoconcentration (Table 3).

We next performed a subgroup analysis, examining the association between sTREM-1 and hemoconcentration in children below and above 9 years of age (Table 4). Of note, the association was of similar magnitude and was statistically significant in both subgroups.

### Elevated sTREM-1 is associated with higher neutrophil count and systemic inflammation

We hypothesized that sTREM-1 would be associated with myeloid cells. Indeed, patients with elevated sTREM-1 exhibited a significantly higher total white blood cell (WBC) count and absolute neutrophil count, whereas the lymphocyte count was similar (Fig 2A-2C). The correlation coefficients between sTREM-1 and WBC, neutrophil, and lymphocyte counts are shown in Table 5.

We further hypothesized that sTREM-1 would be associated with systemic inflammation. Elevated TNF (≥5 pg/mL) was more common in the group with high sTREM-1 (33% versus 14%, P=0.0014, Fig 2D). The correlation coefficient between sTREM-1 and TNF is shown in Table 5.

### Elevated sTREM-1 is associated with markers of endothelial activation and hemoconcentration

We next investigated possible mechanisms by which sTREM-1 may lead to hemoconcentration. Endothelial activation markers Ang2, sFlt-1, and sVCAM-1 were higher in patients with elevated sTREM-1 (Fig 3). The correlation coefficients between sTREM-1 and markers of endothelial activation are shown in Table 5. These markers were also higher in patients with hemoconcentration (p<0.0001 for all three comparisons).

Levels of sTREM-1, TNF, Ang2, sFlt-1, and sVCAM-1 were summarized, stratified by disease severity category (Table 6), and by NS1 antigen levels (Table 7). Of note, detectable NS1 antigenemia was associated with elevated sTREM-1 (Table 7).

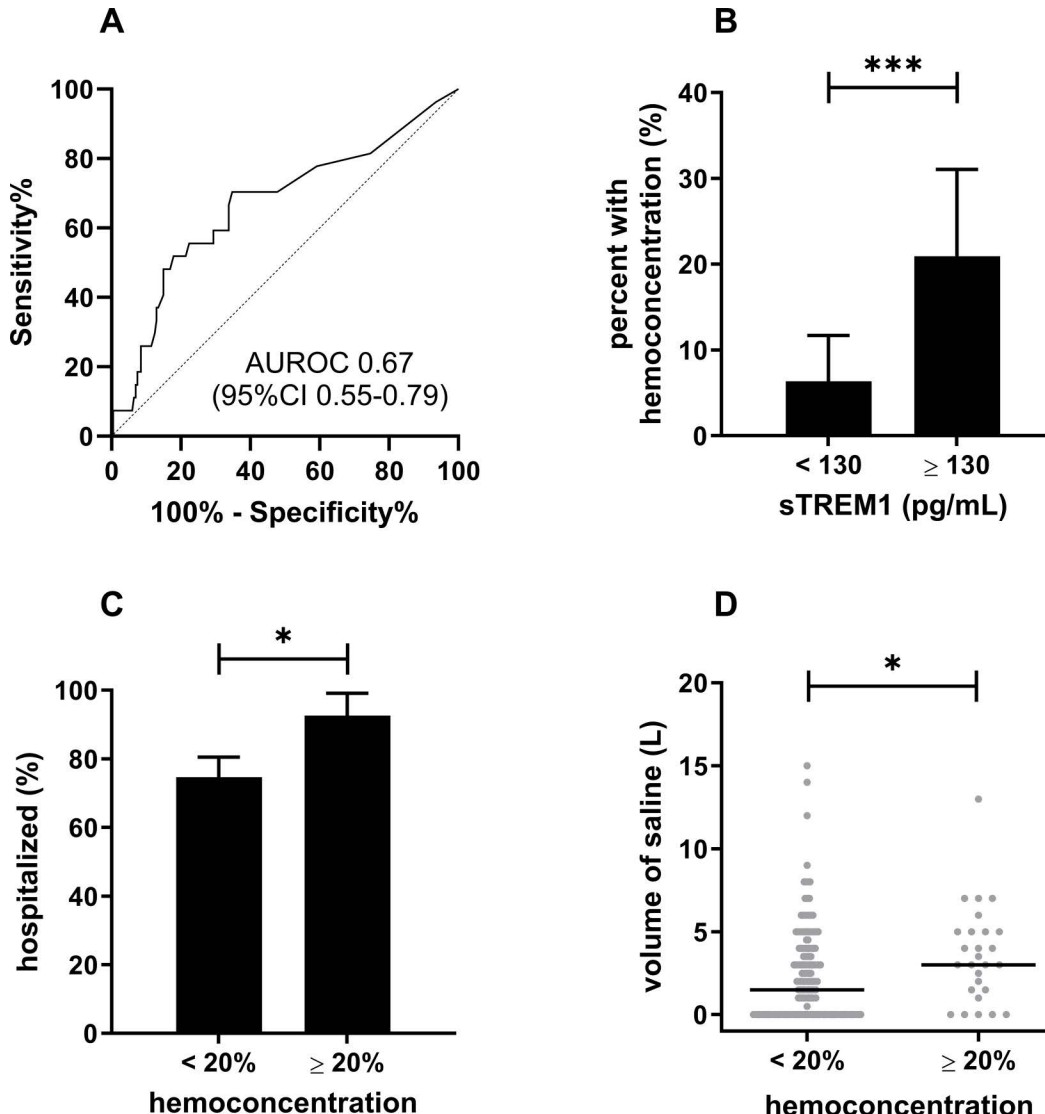

**Fig 1. Elevated sTREM-1 at outpatient clinic presentation was associated with hemoconcentration.** Receiver operator characteristic curve for sTREM-1 used to discriminate between patients with and without hemoconcentration **(A)**. Elevated sTREM-1 (≥130 pg/mL) was associated with hemoconcentration **(B)**. In turn, hemoconcentration was associated with hospitalization **(C)** and higher volume of intravenous saline for supportive management **(D)**. *$p < 0.05$; ***$p < 0.001$.

## Discussion

In this prospective cohort study, we demonstrated that sTREM-1 is associated with hemoconcentration, systemic inflammatory response, and endothelial activation. Thus, sTREM-1 may be a clinically informative indicator of plasma leakage and disease severity. Our results confirm and extend those of a previous study that identified sTREM-1 as one of a combination of markers that was associated with progression to moderate or severe dengue [14]. Our study is noteworthy for the inclusion of outpatients early in the disease course with uncertain disease trajectory. This group is representative of the largest number of DENV cases in which clinicians are faced with challenging decisions about admission versus outpatient management.

**Table 3. Univariable and multivariable logistic regression models for variables associated with hemoconcentration.**

| | Univariable | | Multivariable | |
|---|---|---|---|---|
| | OR (95%CI) | P-value | aOR (95%CI) | P-value |
| sTREM-1 (per 100 pg/mL increase)[a] | 2.3 (1.4-3.8) | 0.0017 | 2.4 (1.4-4.1) | **0.00096** |
| Female sex | 1.8 (0.79-4.0) | 0.17 | 1.8 (0.78-4.2) | 0.17 |
| Time from symptom onset (per day) | 0.92 (0.70-1.2) | 0.53 | 0.86 (0.64-1.1) | 0.29 |

[a]The odds ratio represents the fold-increase in the odds of hemoconcentration per unit change in sTREM-1 levels. One unit change in sTREM-1was defined as 100 pg/mL.

**Table 4. Subgroup analysis of sTREM-1, hemoconcentration, and their association, stratified by patient age.**

| Subgroup | Elevated sTREM-1, n (%) | Hemoconcentration, n (%) | OR (95%CI) | P-value |
|---|---|---|---|---|
| <9 years (N = 118) | 44 (35) | 15 (13) | 3.4 (1.1-11) | **0.039** |
| ≥9 years (N = 126) | 48 (41) | 12 (11) | 4.8 (1.3-24) | **0.026** |

Our primary analysis showed that elevated sTREM-1 (≥130 pg/mL) was associated with a 3.8-fold higher odds (95%CI 1.6-10) of hemoconcentration, a sign of plasma leakage. All cases of hemoconcentration were present at the time of presentation and resolved thereafter, suggesting that vascular leak was most severe at the initial encounter. The association between sTREM-1 and hemoconcentration was robust to adjustment for clinical covariates (Table 3) and remained statistically significant in subgroups stratified by age (Table 4).

Consistent with its known myeloid source, elevated sTREM-1 was associated with higher absolute neutrophil count in our study (Fig 2B). In an experimental study, co-incubation of cultured human neutrophils with DENV-2 resulted in increased surface expression of TREM-1 [13]. In a clinical observational study from Mexico, TREM-1 appeared to be released from the neutrophil cell surface early in the course of DENV infection [14]. Together, these observations suggest that neutrophils contribute to the circulating pool of sTREM-1 through shedding of TREM-1 from the cell membrane. We also found that elevated sTREM-1 was associated with elevated TNF [17,18]. Previous authors have reported an association between sTREM-1 and inflammatory markers results in DENV infection [15] and other conditions [19]. In addition, sTREM-1 was higher in patients with detectable NS1 antigen (Table 7), suggesting that higher viral load may be a driver of neutrophil activation.

Neutrophil activation may contribute to endothelial activation and hyperpermeability [14,20]. We found that sTREM-1 levels correlated with Ang2, sVCAM1, and sFlt1 (Fig 2D-2F). Previous studies have found an association between elevated levels of these endothelial-specific markers and vascular leak in dengue [21,22]. Clinically apparent plasma leakage (hemoconcentration) and fluid replacement therapy were also associated with higher sTREM-1 in our cohort. Our findings are consistent with a previous study that showed a positive correlation between sTREM-1 and Ang-2 [14].

Our study has several limitations. We did not have information on the DENV serotypes or on past infection of heterologous serotype. Only two patients developed severe dengue and there were no fatalities, limiting the generalizability of our findings to more severe cases. Because it was observational, our study lacked the ability to draw causal inferences. The lower limits of detection for sTREM-1 (45 pg/mL) and TNF (5 pg/mL) in our assay resulted in undetectable values in some patients. A more sensitive assay would be required for accurate quantification of sTREM and TNF levels in DENV infection. To address this limitation, we analyzed sTREM-1 and TNF as binary variables, allowing their unambiguous classification into "high" and "low" levels. Our study focused on DENV infection; however, sTREM-1 is a host response marker that may be clinically informative in other febrile illnesses. Further studies are warranted to examine the utility of sTREM-1 in

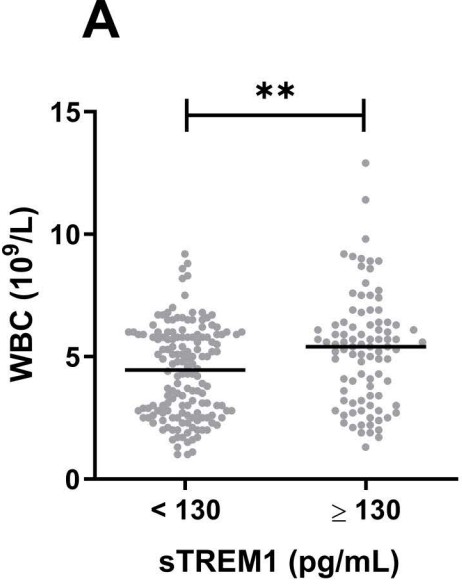

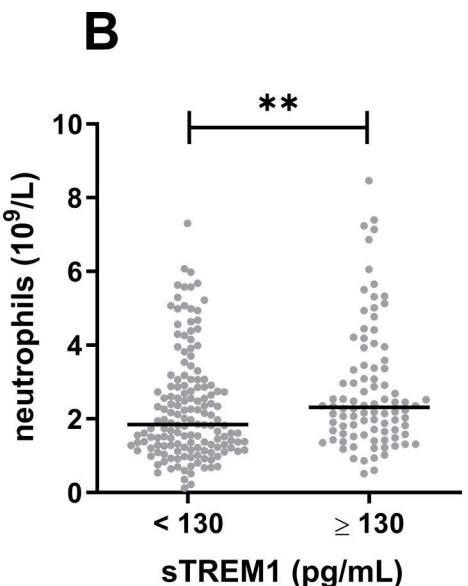

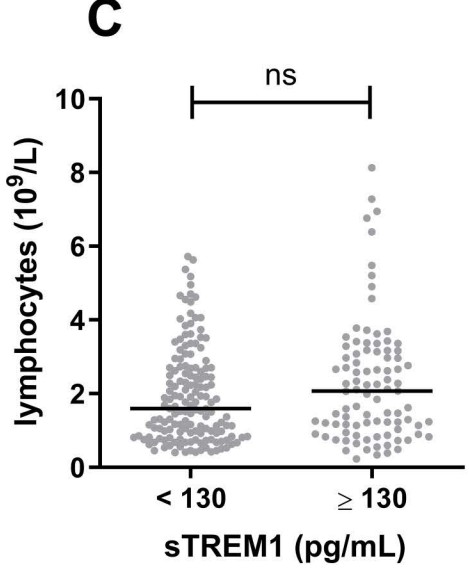

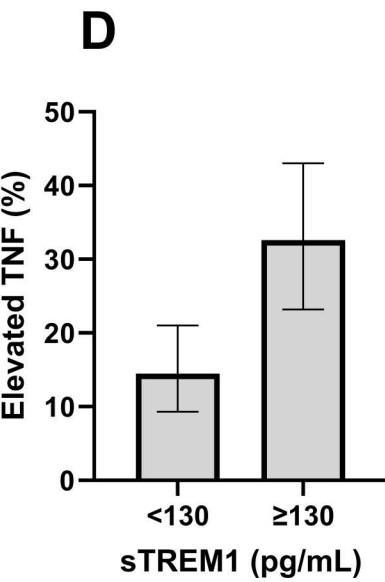

**Fig 2. Associations of sTREM-1 with leukocyte counts and tumor necrosis factor (TNF).** Elevated sTREM-1 (≥130 pg/mL) was associated with higher total white blood cell (WBC) count (**A**) and absolute neutrophil count (**B**), but not lymphocyte count (**C**). **D.** The proportion of patients with elevated TNF (>5 pg/mL) was higher in patients who also had elevated sTREM-1. **\*\***$p < 0.01$; ns, non-significant.

**Table 5. Correlation of sTREM-1 with cellular and molecular markers of inflammation and endothelial activation.**

|  | Correlation coefficient (Kendall's $\tau_B$) | P-value |
|---|---|---|
| WBC ($10^9$/L) | 0.16 | 0.00047 |
| Neutrophil count ($10^9$/L) | 0.14 | 0.0036 |
| Lymphocyte count ($10^9$/L) | 0.09 | 0.052 |
| TNF (pg/mL) | 0.18 | 0.00082 |
| Ang2 (ng/mL) | 0.16 | 0.00078 |
| sFlt-1 (pg/mL) | 0.2 | <0.0001 |
| sVCAM-1 (mg/mL) | 0.14 | 0.0024 |

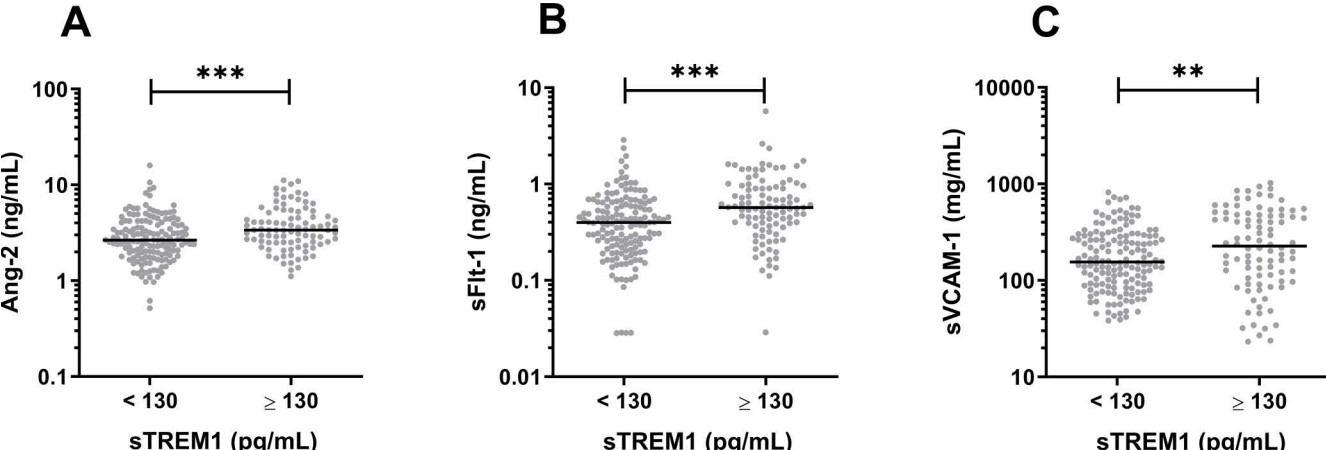

**Fig 3. Associations of sTREM-1 with markers of endothelial activation.** Elevated sTREM-1 (≥130 pg/mL) was associated with elevated markers of endothelial activation Ang-2 **(A)**, sFlt-1 **(B)**, and sVCAM-1 **(C)**. **$p < 0.01$;***$p < 0.001$.

**Table 6. Levels of sTREM-1, TNF, and markers of endothelial activation, stratified by disease severity category.**

|  | Uncomplicated (N = 63)[a] | Warning signs (N = 179) | Severe[a] (N = 2) |
|---|---|---|---|
| sTREM-1 (pg/mL) | 45 (45-140) | 73 (45-150) | 130, 160 |
| Elevated sTREM-1[b], n (%) | 20 (32) | 70 (39) | 2 (100) |
| TNF (pg/mL), median (range) | 5 (5-22) | 5 (5-37) | 5, 15 |
| Elevated TNF[c], n (%) | 11 (17) | 40 (22) | 1 (50) |
| Ang-2 (ng/mL) | 2.5 (1.7-3.7) | 3.1 (2.3-4.3) | 3.4, 4.5 |
| sFlt1 (pg/mL) | 310 (190-530) | 500 (300-840) | 390, 570 |
| sVCAM-1 (mg/mL) | 140 (89-250) | 200 (100-400) | 180, 270 |

Data shown represent median (IQR) unless otherwise indicated.

[a]Only two patients had severe dengue. Individual values are given.

[b]Elevated sTREM-1 was defined as ≥ 130 pg/mL.

[c]Elevated TNF was defined as > 5 pg/mL.

**Table 7. Levels of sTREM-1, TNF, and markers of endothelial activation, stratified by DENV nonstructural protein 1 (NS1) level.**

| | NS1 detectable (N = 189) | NS1 not detectable (N = 55) | P-value |
|---|---|---|---|
| sTREM-1 (pg/mL) | 45 (45-140) | 140 (45-170) | **0.0015** |
| Elevated sTREM-1[a], n (%) | 62 (33) | 30 (55) | **0.0056** |
| TNF (pg/mL), median (range) | 5 (5-37) | 5 (5-27) | **0.038** |
| Elevated TNF[b], n (%) | 46 (24) | 6 (11) | 0.051 |
| Ang-2 (ng/mL) | 3.0 (2.2-4.2) | 2.9 (1.8-4.1) | 0.46 |
| sFlt1 (pg/mL) | 450 (270-720) | 480 (190-640) | 0.42 |
| sVCAM-1 (mg/mL) | 170 (96-310) | 240 (91-380) | 0.60 |

Data shown represent median (IQR) unless otherwise indicated.

[a]Elevated sTREM-1 was defined as ≥ 130 pg/mL.

[b]Elevated TNF was defined as > 5 pg/mL.

other infections. Furthermore, our study did not include healthy controls. sTREM-1 levels should be assessed in healthy individuals without acute infection in order to establish a reference range for this marker. Future experimental research on TREM-1 (e.g., *in vitro* or animal models) would be useful to dissect the underlying mechanisms of the sTREM-1-DENV interaction.

In conclusion, among patients presenting to an outpatient clinic with DENV infection, sTREM-1 was predictive of hemoconcentration, an indicator of vascular leak associated with adverse clinical events (hospitalization, intravenous fluid resuscitation). sTREM-1 may be a marker of pathological processes leading to vascular permeability, including inflammation, neutrophil activation, and endothelial activation. Our findings may have clinical implications. For instance, dampening excessive neutrophil activation may be a therapeutic strategy. Integrating sTREM-1 into rapid diagnostic tests could facilitate the identification of at-risk patients and those who can be safely managed as outpatients. Future exploration is warranted to better assess the role of sTREM-1 in forecasting disease severity and improving outcomes in DENV infection.

## Supporting information

**S1 Data. Raw data used for analysis.** File name: sTREM1 Data.xlsx. File type: Microsoft Excel.
(XLSX)

## Author contributions

**Conceptualization:** Michelle Ngai, Michael T. Hawkes.

**Data curation:** Maria Salome Siose Painaga.

**Formal analysis:** Jiayi Yang, Michael T. Hawkes.

**Funding acquisition:** Ashley Roberts, Kevin C. Kain.

**Methodology:** Kevin C. Kain, Michael T. Hawkes.

**Project administration:** Maria Salome Siose Painaga, James Yared Gaite.

**Supervision:** James Yared Gaite.

**Writing – original draft:** Jiayi Yang.

**Writing – review & editing:** Hridesh Mishra, Michelle Ngai, Vanessa Tran, Maria Salome Siose Painaga, James Yared Gaite, Ashley Roberts, Kevin C. Kain, Michael T. Hawkes.

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
