## [Decision Letter · Decision Letter 0]

28 Jan 2025

PNTD-D-24-01496

Soluble triggering receptor expressed on myeloid cells 1 predicts vascular leak in children and young adults with dengue virus infection in the Philippines

Dear Dr. Hawkes,

Thank you for submitting your manuscript to PLOS Neglected Tropical Diseases. After careful consideration, we feel that it has merit but does not fully meet PLOS Neglected Tropical Diseases's publication criteria as it currently stands. Therefore, we invite you to submit a revised version of the manuscript that addresses the points raised during the review process.

Please submit your revised manuscript within 60 days Mar 29 2025 11:59PM. If you will need more time than this to complete your revisions, please reply to this message or contact the journal office at plosntds@plos.org. Please include the following items when submitting your revised manuscript:

We look forward to receiving your revised manuscript.

Kind regards,

Antonio Mas

Academic Editor

Andrea Marzi

Section Editor

Shaden Kamhawi

co-Editor-in-Chief

Paul Brindley

co-Editor-in-Chief

**Journal Requirements:**

1) Please provide an Author Summary. This should appear in your manuscript between the Abstract (if applicable) and the Introduction, and should be 150-200 words long. The aim should be to make your findings accessible to a wide audience that includes both scientists and non-scientists. Sample summaries can be found on our website under Submission Guidelines:

- ® on page: 5.

5) We note that your Data Availability Statement is currently as follows: "Data cannot be shared publicly because of patient confidentiality. Data are available from the corresponding author for researchers who meet the criteria for access to confidential data.". Please confirm at this time whether or not your submission contains all raw data required to replicate the results of your study. Authors must share the “minimal data set” for their submission. PLOS defines the minimal data set to consist of the data required to replicate all study findings reported in the article, as well as related metadata and methods (https://journals.plos.org/plosone/s/data-availability#loc-minimal-data-set-definition).

- The points extracted from images for analysis..

**Reviewers' Comments:**

Reviewer's Responses to Questions

**Key Review Criteria Required for Acceptance?**

**Methods**

-Are the objectives of the study clearly articulated with a clear testable hypothesis stated?

-Is the study design appropriate to address the stated objectives?

-Is the population clearly described and appropriate for the hypothesis being tested?

-Is the sample size sufficient to ensure adequate power to address the hypothesis being tested?

-Were correct statistical analysis used to support conclusions?

-Are there concerns about ethical or regulatory requirements being met?

Reviewer #1: - The authors have written the Study Procedures at Methods, describing Venipuncture blood was collected for complete blood count, nevertheless, the chronological point during the infection course when the venipuncture was realised is not defined. Furthermore, at the last paragraph of the Introduction, the authors have written Our primary objective was to investigate the association between sTREM-1 and subsequent hemoconcentration, however, it is not explained when they have calculated the hemoconcentration. I think that including a further description of the chronology of the samples extraction should be essential for a better understanding of the whole manuscript.

- The authors have indicated at Study Procedures that Participants were followed-up after resolution of the acute illness (day 14-21 after presentation) to review and document their clinical course, including hospitalization and intravenous fluid management, in spite of that, they have not included in Table 1 or in another table which characteristics or symptons they collected at that point. Moreover, they have not included data about hospitalization at Table 1, unless that Warning signs and Severe dengue indicated hospitalization, even so, I think that could be appreciated to add or clarify all this information.

- As a minor recommendation: The authors should include the number of patients included at Study Design and Participants (Methods).

Reviewer #2: The objectives of the study are well articulated and support the hypothesis stated by the authors. However, there are some issues to address before considering it for publication.

Reviewer #3: There are several serious flaws in the methods and analysis that must be addressed; see full review below and attached.

**Results**

-Does the analysis presented match the analysis plan?

-Are the results clearly and completely presented?

-Are the figures (Tables, Images) of sufficient quality for clarity?

Reviewer #1: I think that the figures designed for the Results are really accurate and show the data in an easy way to understand. However, I suggest to develop somewhat the results, showing the marker values, in order to help other researchers in their studies of endothelial activation.

Reviewer #2: The data presented matched the anlaysis plan and are well presented. However, some data still neeed to be clarified before considering it for publication.

Reviewer #3: 1. Predictive value of sTREM-1:

The manuscript lacks clarity on the time points for sample collection and outcome development. These details are critical for establishing the predictive value of sTREM-1 levels, as claimed by the authors in the title. To claim prediction, it is imperative to provide evidence that the exposure (sTREM-1 levels) was present at a date before the outcome (hemoconcentration) was measured. Otherwise, the study shows association not prediction.

2. Data analysis:

a. The authors should conduct a logistic regression analysis to assess the relationship between sTREM-1 levels (quantitative) and the odds of developing hemoconcentration. This would strengthen the evidence for the predictive role of sTREM-1. Based on the data on Table 1, it appears that sex is a potential confounder in the analysis. The authors should adjust by this variable in a multivariate analysis.

b. Given that based on what is currently reported in the study, it is difficult to establish if the exposure was present before the development of the outcome, a more appropriate measurement of the strength of association would be the Odds Ratio, instead of the Relative Risk.

3. Variable correlations:

The authors should show the correlations between white blood cell count, neutrophils, platelet, lymphocytes, Ang-2, sVCAM-1, and sFlt-1 with the quantitative values of sTREM-1 (instead of its categorial values <130pg/mL, ≥130pg/mL). These analyses would provide a more comprehensive understanding of the data.

**Conclusions**

-Are the conclusions supported by the data presented?

-Are the limitations of analysis clearly described?

-Do the authors discuss how these data can be helpful to advance our understanding of the topic under study?

-Is public health relevance addressed?

Reviewer #1: - The conclusions are supported by the fact that the sTREM-1 high levels are significantly associated with the hemoconcentration and the high levels of inflammation markers (WBC, neutrophils, TNF, Ang2, sFlt-1, sVCAM-1). It has allowed the authors to propose sTREM-1 as a marker of severe progression in dengue virus infection and that could facilitate the identification of those patients at risk. However, the title of the manuscript may not seem the most suitable with the conclusions presented at the end of the Discussion ("sTREM-1 was predictive of hemoconcentration"). Taking this sentece as the main conclusion, I consider that it could be interesting to draw the title up again from another point of view.

- The limitations of analysis are clearly described at the Discussion.

- Furthermore, these findings could have clinical implications as they mentioned in the Discussion: "Integrating sTREM-1 into rapid diagnostic tests could facilitate the identification of at-risk patients and those who can be safely managed as outpatients".

Reviewer #2: Conclusions are mostly sported by the presented data; however, there are some issues to be resolved as depicted below.

Reviewer #3: As presented, the claim of prediction cannot be substantiated; please see full review below and attached.

**Editorial and Data Presentation Modifications?**

Reviewer #1: The authors have analysed the values from children and young patients, but it seems that they have not analysed those data separately. From my point of view, I think that it could be useful to do that analysis in order to observe any differences in the correlation of the markers between children and young patients, or even between children and teenagers with young patients, due to the fact that the immune system is developing until the adolescence.

In addition:

- The authors do not mention where they have realised the Hematologic parameters included in Table , I may recommend them to do it.

- The authors have described the association between sTREM-1 and TNF but they have not included a figure with that information, I think that it would be appealing to add it.

Reviewer #2: Please consider including some additional info on days of sample collection and day of onset on symptoms for dengue disease, disease severity classification based on the WHO guidelines for dengue manifestaions. Please read summary and general comments included below.

Reviewer #3: 1. Ensure compliance with the STROBE (Strengthening the Reporting of Observational Studies in Epidemiology) guidelines.

2. The subheading: “Elevated sTREM-1 is associated with endothelial activation” should be to a more precise statement “Elevated sTREM-1 is associated with markers of endothelial activation”

**Summary and General Comments**

Reviewer #1: One of the strengths of this study is the large cohort which includes molecular and clinical collected data. On the other hand, the authors have made an introduction of dengue fever and its incidence, but the context explanation of the disease around the world looks a bit short. It could be convenient to extend partially the introduction in order to justify the study and to highlight its originality, maybe it could be useful to add some references in that part. I may recommend to add as a reference the following review:

Clinical predictors of severe dengue: a systematic review and meta-analysis – Tsheten Tsheten, Archie C.A. Clements, Darren J. Gray, Ripon K. Adhikary, Luis Furuya-Kanamori and Kinley Wangdi. Infect Dis Poverty (2021).

Reviewer #2: Findings biomarkers that predict the development of moderate or severe dengue disease is a critical step to improve dengue prognosis and patient treatment. Please address all the following questions/comments/suggestions depicted below:

Major(s)

1. Why not samples obtained from other febrile illnesses or healthy individuals were included as part of the analyses? Please clarify this point as comparative analyses for identifying infectious disease biomarkers must include other febrile infections and also healthy individuals.

2. According to the WHO guidelines for classifying dengue disease manifestations, how could the authors classify their study population? This is an important data that must be included as part of the analyses as it seems that the levels of TREM1 can be used as a clinically informative prognostic marker.

3. Having that in mind, can the authors stratify how the levels of soluble TREM1 and the others inflammatory markers changed based on the dengue disease classification and the level of severity identified for all the dengue cases?

4. Were the levels of NS1 measured? If so, how do NS1 levels correlate with the levels of TREM1 and the other inflammatory markers? NS1 can be a viral determinant of severity associated to endothelial dysfunction leading to increased vascular leak, a hallmark of severe dengue disease.

5. Hemoconcentration refers to a condition where the concentration of blood components like red blood cells increases within the blood due to a decrease in plasma volume, thereby leading to a higher concentration of all plasma components, including proteins, electrolytes, and other substances present in the plasma; essentially, when plasma volume decreases, the concentration of everything within that plasma becomes more concentrated. In this sense, did not the authors expected to find also increased concentrations of TREM1 in the more hemoconcentrated plasmas? In other words, how can the authors exclude the fact that less plasma volume in more hemoconcentrated samples led to higher concentration of TREM-1 and other soluble markers measured in this study?

6. Following the previous question, did authors considered including the days post-onset of symptoms as part of the analyses on the TREM1 levels? Can that influence the levels of TREM1 detected in plasmas?

Reviewer #3: This manuscript by Yang et al. explores the relationship between the soluble form of Triggering Receptor Expressed on Myeloid Cells 1 (sTREM-1) and hemoconcentration, a hallmark of vascular leakage in dengue. Previous studies have investigated sTREM-1 as a potential predictor of severe dengue, particularly when combined with other inflammatory markers.

In their study, Yang et al. analyzed 244 children and young adults (aged 1–26 years), measuring sTREM-1 levels using a multiplex Luminex® assay. They assessed the correlation between sTREM-1 levels and the development of hemoconcentration, defined as an increase of >20% from baseline hematocrit. Additionally, the authors examined the association between hemoconcentration and the risk of hospitalization, as well as the need for fluid resuscitation. Their findings indicate a significant association between elevated sTREM-1 levels and hemoconcentration, and between hemoconcentration and increased hospitalization risk and higher fluid management requirements.

However, the study's main limitation lies in the lack of evidence demonstrating that elevated sTREM-1 levels precede the onset of hemoconcentration. This omission undermines the primary claim in the title, “Soluble Triggering Receptor Expressed on Myeloid Cells 1 Predicts Vascular Leak in Children and Young Adults with Dengue Virus Infection,” as the predictive role of sTREM-1 cannot be conclusively established without temporal data showing that the exposure (sTREM-1 levels) was present before the outcome (hemoconcentration). Moreover, the authors show that sTREM-1 levels were also associated with sex (Table 1), indicating the need to address potential confounders in a multivariate analysis.

Major comments:

1. Predictive value of sTREM-1:

The manuscript lacks clarity on the time points for sample collection and outcome development. These details are critical for establishing the predictive value of sTREM-1 levels, as claimed by the authors in the title. To claim prediction, it is imperative to provide evidence that the exposure (sTREM-1 levels) was present at a date before the outcome (hemoconcentration) was measured. Otherwise, the study shows association not prediction.

2. Data analysis:

a. The authors should conduct a logistic regression analysis to assess the relationship between sTREM-1 levels (quantitative) and the odds of developing hemoconcentration. This would strengthen the evidence for the predictive role of sTREM-1. Based on the data on Table 1, it appears that sex is a potential confounder in the analysis. The authors should adjust by this variable in a multivariate analysis.

b. Given that based on what is currently reported in the study, it is difficult to establish if the exposure was present before the development of the outcome, a more appropriate measurement of the strength of association would be the Odds Ratio, instead of the Relative Risk.

3. Variable correlations:

The authors should show the correlations between white blood cell count, neutrophils, platelet, lymphocytes, Ang-2, sVCAM-1, and sFlt-1 with the quantitative values of sTREM-1 (instead of its categorial values <130pg/mL, ≥130pg/mL). These analyses would provide a more comprehensive understanding of the data.

Minor comments:

1. Ensure compliance with the STROBE (Strengthening the Reporting of Observational Studies in Epidemiology) guidelines.

2. The subheading: “Elevated sTREM-1 is associated with endothelial activation” should be to a more precise statement “Elevated sTREM-1 is associated with markers of endothelial activation”

PLOS authors have the option to publish the peer review history of their article (what does this mean? ). If published, this will include your full peer review and any attached files.

**Do you want your identity to be public for this peer review?** For information about this choice, including consent withdrawal, please see our Privacy Policy .

Reviewer #1: No

Reviewer #2: No

Reviewer #3: No

**Figure resubmission:**
---

## [Decision Letter · Decision Letter 1]

25 Apr 2025

Dear Dr. Hawkes,

We are pleased to inform you that your manuscript 'Soluble triggering receptor expressed on myeloid cells 1 is associated with hemoconcentration and endothelial activation in children and young adults with dengue virus infection in the Philippines' has been provisionally accepted for publication in PLOS Neglected Tropical Diseases.

Best regards,

Antonio Mas

Academic Editor

Andrea Marzi

Section Editor

Shaden Kamhawi

co-Editor-in-Chief

Paul Brindley

co-Editor-in-Chief

Reviewer's Responses to Questions

**Key Review Criteria Required for Acceptance?**

**Methods**

-Are the objectives of the study clearly articulated with a clear testable hypothesis stated?

-Is the study design appropriate to address the stated objectives?

-Is the population clearly described and appropriate for the hypothesis being tested?

-Is the sample size sufficient to ensure adequate power to address the hypothesis being tested?

-Were correct statistical analysis used to support conclusions?

-Are there concerns about ethical or regulatory requirements being met?

Reviewer #1: The authors have written the number of patients in 'Study Design' and 'Participants' as I mentioned in the review. Also, the authors have included a new table (Table 2) and a paragraph in the 'Results' that clarifies the follow-up of the patients and facilitates the understanding of the characteristics of the cohort.

Regarding the chronology of the sample extraction, I think that now the study procedure is clearer than before thanks to the new 'Study Procedures' that the authors have written in order to explain with more detail the methods.

Reviewer #2: After this second round of revision, the objectives of the study are now clearly articulated with a clear testable hypothesis stated.

**Results**

-Does the analysis presented match the analysis plan?

-Are the results clearly and completely presented?

-Are the figures (Tables, Images) of sufficient quality for clarity?

Reviewer #1: In my opinion, the fact that the authors have included the values for endothelial markers it will be really useful for other researchers in order to compare their own results with these ones.

Reviewer #2: After this second round of revision, results, figures and tables are clearly and completely presented.

**Conclusions**

-Are the conclusions supported by the data presented?

-Are the limitations of analysis clearly described?

-Do the authors discuss how these data can be helpful to advance our understanding of the topic under study?

-Is public health relevance addressed?

Reviewer #1: I think that the new title emphasizes better the main ideas of the research and points out the conclusions that the authors have obtained.

Reviewer #2: After this second round of revision, conclusions are better supported by the data presented, including the limitations of the study.

**Editorial and Data Presentation Modifications?**

Reviewer #1: I really appreciate that the authors have followed my recommendation about analyse the data separately. Although the new analysis does not show significant difference between both groups, I think it is interesting that the authors have included this information.

On the other hand, I think that the addition of the Figure 2D is really informative and it improves the quality of the results exposition.

Reviewer #2: (No Response)

**Summary and General Comments**

Reviewer #1: Regarding the introduction, I think that all the new information that the authors have included is really useful to create a context about the disease and how important the impact of this study could be. Furthermore, the more extended description of the symptoms makes easier to understand the clinic of the disease for the reader.

Reviewer #2: (No Response)

PLOS authors have the option to publish the peer review history of their article (what does this mean? ). If published, this will include your full peer review and any attached files.

**Do you want your identity to be public for this peer review?** For information about this choice, including consent withdrawal, please see our Privacy Policy .

Reviewer #1: **Yes: ** Lourdes Arias-Salazar

Reviewer #2: No

---

## [Editor Report · Acceptance letter]

Dear Dr. Hawkes,

We are delighted to inform you that your manuscript, "Soluble triggering receptor expressed on myeloid cells 1 is associated with hemoconcentration and endothelial activation in children and young adults with dengue virus infection in the Philippines," has been formally accepted for publication in PLOS Neglected Tropical Diseases.

Best regards,

Shaden Kamhawi

co-Editor-in-Chief

Paul Brindley

co-Editor-in-Chief
